# The Impact of Spatial Delineation on the Assessment of Species Recovery Outcomes

**Molly K. Grace [1,\*], Haluk Resit Akçakaya [2], Elizabeth L. Bennett [3], Michael J. W. Boyle [4,5], Craig Hilton-Taylor [6], Michael Hoffmann [7], Daniel Money [4], Ana Prohaska [4,8], Rebecca Young [1,9], Richard Young [9] and Barney Long [10]**

1   Department of Biology, University of Oxford, Oxford OX1 3SZ, UK
2   Department of Ecology and Evolution, Stony Brook University, Stony Brook, NY 11794, USA
3   Wildlife Conservation Society, Bronx, NY 10460, USA
4   Department of Zoology, University of Cambridge, Cambridge CB2 3EJ, UK
5   School of Biological Sciences, University of Hong Kong, Hong Kong 999077, China
6   IUCN Red List Unit, Cambridge CB2 3QZ, UK
7   Conservation Programmes, Zoological Society of London, London NW1 4RY, UK
8   Lundbeck Foundation GeoGenetics Centre, GLOBE Institute, University of Copenhagen,
    1350 Copenhagen, Denmark
9   Durrell Wildlife Conservation Trust, Trinity JE3 5BP, Jersey, Channel Islands, UK
10  Re:wild, Austin, TX 78767, USA
\*   Correspondence: molly.grace@wadham.ox.ac.uk

**Abstract:** In 2021, the International Union for Conservation of Nature (IUCN) introduced a novel method for assessing species recovery and conservation impact: the IUCN Green Status of Species. The Green Status standardizes recovery using a metric called the Green Score, which ranges from 0% to 100%. This study focuses on one crucial step in the Green Status method—the division of a species' range into so-called "spatial units"—and evaluates whether different approaches for delineating spatial units affect the outcome of the assessment (i.e., the Green Score). We compared Green Scores generated using biologically based spatial units (the recommended method) to Green Scores generated using ecologically based or country-based spatial units for 29 species of birds and mammals in Europe. We found that while spatial units delineated using ecoregions and countries (fine-scale) produced greater average numbers of spatial units and significantly lower average Green Scores than biologically based spatial units, coarse-scale spatial units delineated using biomes and countries above a range proportion threshold did not differ significantly from biologically based results for average spatial unit number or average Green Score. However, case studies focusing on results for individual species (rather than a group average) showed that, depending on characteristics of the species' distribution, even these coarse-scale delineations of ecological or country spatial units often over- or under-predict the Green Score compared to biologically based spatial units. We discuss cases in which the use of ecologically based or country-based spatial units is recommended or discouraged, in hopes that our results will strengthen the new Green Status framework and ensure consistency in application.

**Keywords:** green status; IUCN; red list; subpopulations; viability

## 1. Introduction

In 2021, the International Union for Conservation of Nature (IUCN) introduced a novel method for assessing species recovery and conservation impact: the IUCN Green Status of Species (hereafter Green Status) [1]. The draft methods for the Green Status assessment [2] were tested with nearly 200 species [3] and adapted as needed to be applicable to any animal, plant, or fungus species. Input from a number of non-academic potential end-users was also incorporated into the final methods [4]. The rigorous development process has resulted in a standardized way to measure species recovery (Box 1), opening



the door to the development of new recovery indicators and targets. However, while the Green Status assessment methods are the result of years of development and testing, there are still areas of uncertainty that need to be addressed to strengthen this new framework and ensure consistency in application. This is due in part to the fact that the methods require assessors to make a number of choices, and key questions remain about how those choices might affect the results. One area where further guidance is needed is the definition and delineation of spatial units—species-specific divisions within which recovery state is assessed (Box 1).

**Box 1.** Calculating recovery using the IUCN Green Status of Species.

The Green Status assesses recovery on a spectrum from 0% to 100%, where 100% represents a Fully Recovered (or Non-Depleted) species, while 0% represents a species that is Extinct in the Wild.

In brief, the Green Status assesses species recovery by asking assessors to define a species' *indigenous range* (i.e., the area occupied by wild populations of the species at the time when humans began to have a major influence on the species' abundance or distribution [5]). This indigenous range is then divided into parts called *spatial units*, and the species' *state* in each spatial unit is assessed. The potential states are: *Absent* (extirpated), *Present*, *Viable (*not threatened with extinction, as evidenced by the IUCN Red List of Threatened Species Criteria [6,7]), and *Functional* (carrying out its ecological roles and interactions). These states are each given a weight (Absent lowest, Functional highest), and the weights in each spatial unit determine an overall *Green Score*—the recovery value that ranges from 0% to 100%— according to Equation (1):

$$G = \frac{\sum_s W_S}{W_F \times N} \times 100 \tag{1}$$

where s is each spatial unit, $W_s$ is the weight assigned to the state in the spatial unit, $W_F$ is the weight of the Functional state, and N is the number of spatial units. Therefore, the denominator is the maximum possible score that would be attained when all spatial units are assessed as Functional.

If a species were Functional in every spatial unit, it would have the highest possible Green Score of 100%; if it were Absent in every spatial unit (Extinct in the Wild), it would have the lowest possible Green Score of 0%. The full Green Status assessment methods, and explanation of the criteria for assigning one of the four states to a spatial unit, can be found in [1,8].

By dividing a species' indigenous range (Box 1) into spatial units, each of which is then assessed in turn, the Green Status accounts for variation in species' status across the range; in other words, if a species is faring well in some areas, but poorly in others, the Green Score will reflect this heterogeneous reality. The determination of indigenous range and the division of this range into spatial units is therefore a critical step in the Green Status method, but the best way to divide a species' range into meaningful spatial units is not always clear.

The Green Status of Species Standard document (hereafter, the Green Status Standard) [1] indicates that biologically based spatial units are the preferred method of delineation. Biologically based spatial units divide the indigenous range based on relevant biological divisions, for example: subspecies, subpopulations, migratory routes (such as flyways), or evolutionarily significant units [9]. Biologically based delineation of spatial units is assumed to be the most relevant way to divide a species' range because these units represent parts of the range that are dynamically, demographically, or evolutionarily separated from each other. Biogeographic barriers or other natural barriers to dispersal can help define geologically or geographically based spatial units, such as watersheds, mountain ranges, islands, and lakes, which serve as a proxy for delineating biologically based

spatial units, if information about population-dynamic, demographic, or genetic divisions is not available.

However, it is not always possible to create spatial units based on biological divisions or proxies, for example for species with continuous distributions, which makes it difficult or impossible to identify independent subpopulations. Even in these cases, there may nonetheless be variation across the range that should be accounted for (e.g., the species is Functional only in pockets, due to variation in conditions across the range, and has been extirpated from other parts). In such cases, the Green Status Standard indicates that spatial units can be delineated in other ways: one suggestion is using ecological divisions, such as spatially explicit ecoregions. Political divisions, such as regions, countries, or states, are also an option if their boundaries delineate areas of similar threatening processes (i.e., "location" as defined for the Red List). In some cases, species population information and Red List Category (used to assign a state in a spatial unit, IUCN 2021) are held at the country level, so for some species they may be seen as a convenient way to identify spatial units for the assessment. As a last resort, if other divisions are not obvious, grid cells can be laid over the indigenous range and used as spatial units. For a full account of recommended spatial unit delineation methods, see [8].

It is unclear how, or if, the choice of one of these "alternative" methods of delineating spatial units over biological divisions would affect the assessment's resulting Green Score. Given that the Green Status assessment is new, this is the optimal time to provide further clarification to guide assessors and help ensure consistency between assessments. In this paper, we explore how delineation of spatial units based on two division methods—ecological divisions (ecoregions) and political divisions (countries)—changes a species' Green Score compared with the Score generated using biologically based spatial units, and provide recommendations to standardize the process.

## 2. Materials and Methods

### 2.1. Dataset

To compare Green Scores generated for a species using these three different methods of delineating spatial units, we required a set of species for which (1) the indigenous range was mapped; (2) the current distribution and status of the species were known; and (3) biologically based spatial units existed, which we could compare with other delineation methods. We decided to use real species rather than modelled (hypothetical) species to ensure our focal set represented realistic variety in characteristics of true species distributions, and to avoid having to arbitrarily decide what counted as a biologically based spatial unit for a modelled species, which might have rendered the exercise less valuable.

In 2013, Deinet et al. published a report called *Wildlife comeback in Europe: The recovery of selected mammal and bird species* [10]. This comprehensive document recorded the recovery story of 18 mammal species and 19 bird species in Europe, pulling together all available information to map the past and current (at the time of the report) distribution of each species. It also reported on biologically based divisions of the species, if known, including population size and trend within each biological unit for many species. Finally, Deinet et al. reported current population size and trend in the countries currently occupied by the species, if known. The maps and information collated in [10] therefore provided an excellent dataset with which to explore the effects of different methods of spatial unit delineation on reported recovery outcomes.

### 2.2. Species Distributions

A key feature of each species account [10] is a map showing (1) species distribution at the time of publication (data from 2010–2013; the "current" distribution); (2) the species distribution in the "recent past" (1949–1980; median year 1955); and (3) the "historical" species distribution (which varied between species but was generally in the 1800s and was never later than 1900). To carry out Green Status assessments for these species, we would

need to know the distribution at the time when humans began having a major influence on the species' population in the wild. The distribution at this time, plus any natural or conservation-related expansion since major impacts started, represents the indigenous range. The description of this "indigenous range" is the first step in the Green Status assessment process (Box 1). However, the geographical scope of [10] is Europe, which has been home to dense human populations for centuries; given this, many species had already undergone significant declines in range by the 1800s, meaning that the "historical" distribution maps were not representative of the species' indigenous range in most cases. Because we were not attempting to conduct actual Green Status assessments for these species, but rather explore how the outcomes of the Green Score calculation (Equation (1)) change based on different ways of producing spatial units, not knowing the indigenous range was not prohibitive to the intended analysis.

We used the species maps representing the "recent past" reported in [10] as baseline distributions (i.e., for all intents and purposes, we equated these to indigenous range) because (1) not all species had a "historical" map and (2) for many species the "recent past" distribution was larger than the "historical" distribution. Given that the "historical" maps show species status in the 1800s–1900s, which was after industrialization but prior to the conservation movement, it is not surprising that these time points, in some cases, represented species' lows. For this reason, the "baseline" distribution for each species was the distribution in the "recent past" plus any areas that were added in the "current" distribution map. Although this does not hamper our ability to assess how characteristics of a baseline distribution affect the calculation of Green Scores, it does mean that while the species-level results presented are useful for the purposes of this analysis, they should not be treated as an actual Green Status of Species assessment.

For analyses, species distributions were converted into raster layers with a pixel size of 1 km$^2$. In cases where the species distribution in the "recent past" was smaller, or included different areas, than the species' "current" distribution, we combined the two shapefiles for the species to create the operational baseline distribution. In total, we were able to generate baseline distributions for 29 species (15 birds and 14 mammals; Table 1).

Baseline distributions were given one of the following categorical descriptors: in "continuous" distributions ($n = 11$ species), >75% of the baseline range was connected in one unbroken distribution; in "semi-continuous" distributions ($n = 9$ species), the connected area was >50% of the baseline range; and other distributions ($n = 9$ species) were considered "discontinuous".

All spatial and statistical analyses were carried out in R version 3.6.1. Prior to analyses, all spatial data were re-projected from Cartesian to Euclidian geometry using the Lambert Azimuthal Equal Area projection centered on Europe at 9° E and 53° N [11].

**Table 1.** Species results, compared against biological spatial units (SUs). Number of spatial units and Green Scores generated using ecologically based and country-based methods are shown as differences (Δ) from biological results. For context, the baseline distribution is characterized as continuous (C), semi-continuous (S), or discontinuous (D). This table shows the highest-order delineation methods: ecological spatial units are based on biomes, and country-based spatial units include countries comprising ≥ 5% of the baseline distribution. Colored rows indicate species for which the same Green Score was obtained using the biological method and both other methods (dark gray) or one other (light gray). Note that these are not the results that would be generated using a full Green Status assessment (see Section 2) and should therefore not be referenced as the Green Status outputs for these species.

| Taxon | Species | Biological SUs | Green Score | ΔSU$_{Eco}$ * | ΔGS$_{Eco}$ * | ΔSU$_{Country}$ * | ΔGS$_{Country}$ * | Baseline Distribution |
|---|---|---|---|---|---|---|---|---|
| Bird | *Anser brachyrhynchus* | 2 | 67% | 1 | 0% | 0 | 0% | D |
| Bird | *Aquila adalberti* | 1 | 33% | 1 | 0% | 0 | 0% | C |
| Bird | *Ciconia ciconia* | 2 | 67% | 3 | 0% | 5 | 0% | C |
| Bird | *Oxyura leucocephala* | 2 | 17% | 2 | 0% | 2 | 0% | C |
| Mammal | *Cervus elaphus* | 4 | 67% | 3 | 0% | 4 | 0% | C |
| Mammal | *Gulo gulo* | 2 | 33% | 3 | 0% | 2 | 0% | C |

| | | | | | | | | |
|---|---|---|---|---|---|---|---|---|
| Mammal | *Sus scrofa* | 5 | 67% | 2 | 0% | 1 | 0% | C |
| Bird | *Aegypius monachus* | 4 | 42% | 0 | 0% | 2 | −14% | S |
| Bird | *Aquila heliaca* | 3 | 44% | 4 | −2% | 0 | 0% | S |
| Bird | *Branta leucopsis* | 3 | 67% | 2 | −13% | 2 | 0% | D |
| Bird | *Cygnus cygnus* | 2 | 67% | 6 | −21% | 1 | 0% | S |
| Bird | *Grus grus* | 3 | 67% | 5 | −8% | 2 | 0% | C |
| Bird | *Gyps fulvus* | 4 | 50% | 0 | 0% | 3 | −2% | S |
| Mammal | *Bison bonasus* | 2 | 33% | 0 | 17% | 3 | 0% | D |
| Mammal | *Capreolus capreolus* | 3 | 67% | 5 | −8% | 3 | 0% | C |
| Mammal | *Castor fiber* | 5 | 67% | 2 | −5% | 2 | 0% | S |
| Bird | *Falco naumanni* | 3 | 67% | 4 | −19% | 2 | −13% | D |
| Bird | *Falco peregrinus* | 1 | 67% | 7 | −13% | 3 | −8% | S |
| Bird | *Gypaetus barbatus* | 10 | 18% | −7 | 15% | −6 | 15% | D |
| Bird | *Haliaeetus albicilla* | 2 | 67% | 7 | −26% | 1 | −11% | C |
| Bird | *Milvus milvus* | 6 | 56% | 0 | −11% | 0 | −28% | S |
| Mammal | *Canis aureus* | 4 | 50% | −1 | 17% | 4 | 4% | C |
| Mammal | *Canis lupus* | 10 | 43% | −3 | 4% | −4 | 12% | C |
| Mammal | *Capra pyrenaica* | 3 | 44% | −1 | 6% | −2 | 23% | D |
| Mammal | *Lynx lynx* | 11 | 45% | −3 | −8% | −8 | 21% | S |
| Mammal | *Lynx pardinus* | 4 | 17% | −2 | 6% | −3 | 17% | D |
| Mammal | *Rupicapra pyrenaica* | 3 | 56% | −1 | −6% | 1 | −6% | D |
| Mammal | *Rupicapra rupicapra* | 5 | 53% | −2 | 13% | 2 | 13% | D |
| Mammal | *Ursus arctos* | 10 | 43% | −3 | 14% | −7 | 23% | S |

*Δ indicates change relative to results based on biological divisions.

### 2.3. Spatial Units

We compared three different methods of dividing the baseline distribution into spatial units: (1) biological divisions (the recommended method), (2) ecological divisions, and (3) countries. To identify biologically based spatial units, we first referred to the species' account in [10], which sometimes reported information at the level of a recognized biological division (e.g., subspecies, subpopulation), including (in some cases) extirpated ones. We determined if the account reported on biological units (initial read-through of accounts, plus double-check searches for the terms "subspecies", "subpopulations", "populations", "flyways") and compiled the information, if provided. We then checked whether these biological units existed in the year corresponding to the defined baseline distribution, as well as comparing the described biological units to the baseline map to evaluate if any areas of the baseline distribution were not covered by the biological units described (because to calculate the Green Score, all of the baseline distribution must be included in the spatial units). We cross-referenced the Red List account of each species, as well as the available literature, to determine how many biological spatial units would have existed in the baseline year and to determine their status in 2013. For a full account of how biological spatial units were defined for each species, including information sources, see Table S1.

For both "alternative" methods of delineating spatial units, we investigated the effects of dividing a species' range based on different spatial scales: firstly, using a finer spatial scale, which generated larger numbers of spatial units ("fine divisions"), and using a coarser spatial scale, which generated fewer ("coarse divisions"). Ecologically based spatial units were delineated using the original version of the Terrestrial Ecoregions of the World framework [12]. We compared the use of ecoregions as spatial units (fine division) to the use of biomes, which represented groups of ecoregions (coarse division). Because ecoregions nest within biomes, no parts of the range were excluded using either method. The country and ecoregion layers were limited to the extent of "Europe" as defined in [10].

We obtained country boundaries from the package "rworldmap" [13] and evaluated country-based spatial units in three ways for comparison. First, we included every country within the baseline distribution as a spatial unit (fine division). Then, we trialed two types of coarser spatial units: one where we only included countries where the country in question represented at least 1% of the species' baseline distribution, and a second type where we only included countries representing at least 5% of the species' baseline distribution. We evaluated these thresholds because, since all spatial units contribute equally to the Green Score (Equation (1)), we wanted to investigate whether using spatial units that varied greatly in size would result in smaller spatial units depressing the Green Score. Applying these 1% and 5% thresholds for including a country as a spatial unit meant that in some cases, part of the baseline distribution, and the individuals projected within it, was not considered in the calculation of these Green Scores (birds: 1% threshold, average 3.4% of baseline distribution excluded, sd = 3.3%; 5% threshold, average 19.3% of baseline distribution excluded, sd = 11.9%; mammals: 1% threshold, average 2.1% of baseline distribution excluded, sd = 3.0%; 5% threshold, average 13.2% of baseline distribution excluded, sd = 16.6%). In actual Green Status of Species assessments, all parts of the baseline distribution must be included within spatial units.

### 2.4. Population Estimates

To calculate the Green Score of a species under the three different methods of delineating spatial units, we had to determine how many individuals occurred in each spatial unit in 2013. For biological spatial units, these numbers were often reported in [10]. We were also often able to find reported population sizes in 2013 by country in [10], but not always, and population data were not always available in every country, especially for species with large distributions. Population-level data were not reported by ecoregion or biome for any species in [10].

Therefore, to estimate the number of individuals in each ecoregion- or country-level spatial unit, we projected population maps over the 2013 maps of species distributions. We estimated the mean density of individuals per km² by dividing the current total number of individuals in Europe reported in [10] by the total number of occupied 1 km² pixels across the current European range for each species. We extracted the number of occupied 1 km² pixels that fell within each defined spatial unit boundary. We used this value to estimate abundance within different spatial units, by multiplying the number of pixels occupied in 2013 by the mean estimated density of individuals per km². Values were then corrected to estimate the number of mature individuals that would be expected given that population size using values derived from the literature (Table S2). This was done because the number of mature individuals is a key piece of information for determining the state in a spatial unit (Box 1; see also next section, "Calculation of Green Score"). In cases where a correction factor to convert total population size to number of mature individuals could not be derived from the literature for a species, we used information from a closely related species as a proxy (Table S2).

Projecting average density to estimate the number of mature individuals within a spatial unit assumes that the density of individuals is uniform across a species' entire range, which may not be true. To check against this assumption for country-level spatial units, we compared the projected number of mature individuals in each country against the values for that country reported in [10], which are based on observation and inference in the field (Table S3). Specifically, we checked whether any country-level spatial units differed enough in number of mature individuals between the projected and reported values that the state assigned to the spatial unit would be different, which would then change the Green Score (Box 1; see also next section, "Calculation of Green Score"). While we could not do this for ecoregions or biomes, because there were no values reported in [10] against which to compare, we assumed that those projections would perform similarly or better, as ecoregions and biomes were usually larger units than countries.

For mammals, we were able to make a total of 150 comparisons between our projected country-level population size data and the data reported in [10] across 16 mammal species (Table S3). Out of these 150 comparisons, 23 (15.3%) showed that our estimation would result in assigning a different state to a country-level spatial unit than the state indicated by using the numbers reported in [10]. For bird species (*n* = 15), there were 193 comparisons and 30 differences (15.5%). Therefore, in approximately 85% of cases, the projected number of mature individuals produced the same state as the reported values. For many species, the differences canceled each other out (equal number of over- and underestimations), but for seven mammal species, these differences resulted in a different Green Score (six species where the projection method produced a higher Green Score than the values reported in [10] would produce, one where it produced a lower Green Score). This was also the case for seven bird species (four higher, three lower; Table S3).

### 2.5. Calculation of Green Score

For each species, we calculated a Green Score for each method of spatial unit delineation. The Green Status assessment method recognizes four states to which a spatial unit can be assigned: Absent, Present, Viable, or Functional (Box 1) [1]. If, in 2013, zero individuals were observed (biological method) or projected (country, ecoregion methods) in a spatial unit, the spatial unit was considered Absent. If there were >0 individuals in a spatial unit, but it did not meet the criteria for Viable, it was considered Present. For a spatial unit to be considered Viable in a true Green Status assessment, assessors effectively undertake a Red List assessment of the species in the spatial unit applying the regional guidelines [7], and if the resulting category is Least Concern, or if it is Near Threatened but the species is not declining in the spatial unit, the spatial unit is considered Viable. The Red List Criteria are based on the link between small and/or declining populations and extinction risk [14] and therefore check against a number of related factors, including the size of the spatial unit, the population size within it, number of mature individuals, fragmentation within the spatial unit, and whether the spatial unit has had past or ongoing decline, among others. An online tool was developed to facilitate the evaluation of spatial units against these criteria for a Green Status assessment [15].

This comprehensive information needed to check against these criteria was often available for biologically based spatial units; indeed, many biologically based spatial units had been assessed using the regional Red List criteria, with the resulting category reported in [10]. However, this information was not available for ecoregion- and country-based spatial units, which were informed by projected population size estimates. Therefore, for the purposes of this study, we used greatly simplified criteria to evaluate whether a spatial unit was Viable or not. We assumed that at the time of "assessment": (i) no spatial units were experiencing continuing decline or extreme fluctuations, (ii) there was no past or expected future decline, (iii) none were considered severely fragmented or very restricted, and the (iv) "rescue effect" did not apply [7]. Therefore, the only relevant criterion against which to check was whether a spatial unit contained >1000 mature individuals; if it did, it was considered Viable. Note that these assumptions hold only for the purposes of our analysis and would not be assumed a priori for any true Green Status assessment.

To determine viability in country- and ecoregion-based divisions of the baseline range, the projected number of mature individuals in each spatial unit was simply checked against this threshold of 1000 mature individuals. Determining whether biological spatial units qualified as Viable was less straightforward. For some species, population sizes for the biological spatial units were reported in [10]; after applying the correction factor to estimate the number of mature individuals (Table S2), it was a simple exercise to apply the threshold of 1000 mature individuals. Where [10] did not report these values, the species' Red List account was referenced to extract information about the populations within biological spatial units; if that did not yield the needed information, a further literature search was undertaken. In rare cases, the only information available for a biological spatial unit was the regional Red List category and triggering criteria. Because our study

used modified criteria, a Red List categorization pulled from the literature is not directly comparable. To correct for this, if that spatial unit was assigned a threatened category due to any criteria other than having fewer than 1000 mature individuals [6], it was considered Viable in our analysis (as under the assumptions of our simplification, none of the other criteria would be triggered). See Table S4 for a full account of how states were assigned to each species' biological spatial units.

The final possible state for a spatial unit is Functional. In a Green Status assessment, functionality is defined for each species by the assessors, so it was not possible to assess whether spatial units were Functional based on the projected data at our disposal.

There were therefore three possible states to which a spatial unit could be assigned: Absent, Present, or Viable. Each state's corresponding weight (Absent = 0, Present = 3, Viable = 6) was used to calculate the Green Score (Box 1, Equation (1)) for the species under the three methods of delineating spatial units. As previously described, the Green Score formula can produce values between 0% and 100%; however, because we did not assign any spatial units as Functional, the highest possible value in our analysis was 66.7%—the value attained if, in 2013, the species was considered Viable in all spatial units (whichever delineation method was used) throughout the entirety of its baseline European distribution.

### 2.6. Analysis

We assessed the average Green Score of each species for the different methods of spatial unit delineation using beta regression, including taxon (bird or mammal) and baseline distribution category (continuous, semi-continuous, discontinuous) as covariates (R package "betareg") [16,17]. In this analysis, we compared the Green Scores resulting from biologically based delineation of spatial units to both the finer-scale ecologically based and country-based spatial unit delineations (all ecoregions, all countries) as well as to the coarser-scale delineations (ecoregions collapsed into biomes, countries representing ≥1% or ≥5% of the baseline distribution). All *p*-values reported are from likelihood ratio tests and $R^2$ values are from pseudo-$R^2$. Reported estimates and standard errors were converted back to the original scales of the independent variables.

A particular concern that has arisen in some initial testing is that subdividing the indigenous range into a greater number of spatial units could have a depressive effect on the Green Score (i.e., it is much less likely for a species to be categorized as Fully Recovered because there are many more spatial units in which a species must be assessed as Functional). To evaluate this, we calculated the difference in scores generated from each method and plotted the differences against the difference in number of spatial units.

While understanding the average differences in Green Score that can result from the choice of spatial unit delineation method is important, understanding the reasons for these changes for an individual species is perhaps more important for the development of practical guidance for assessors choosing which delineation method to use. We selected species where Green Scores and number of spatial units were either very similar or very different, based on the delineation method used, as case studies to identify characteristics that could be used to develop guidance. These characteristics included baseline distribution size and whether that distribution was continuous or not; in addition, for both the finer-scale and coarser-scale ecologically based and country-based spatial unit delineations, we evaluated how many of the spatial units generated by those methods could ever actually hold a minimum viable population, based on the size of the spatial unit and the density of mature individuals per km² estimated for that species.

### 3. Results

*3.1. Average Changes in Number of Spatial Units Based on Delineation Method*

The number of spatial units generated by the different methods of delineating spatial units also differed between the use of fine and coarse divisions for assigning country-based and ecologically based spatial units. When using fine divisions, species ranges were divided into many more spatial units, on average, using country-based and ecologically based divisions than biologically based divisions (Figure 1). When using coarser divisions, country-based and ecologically based methods of delineating spatial units produced similar average numbers of spatial units to biologically based methods, with the exception of country-based spatial units including all countries comprising at least 1% of the baseline range (Figure 1).

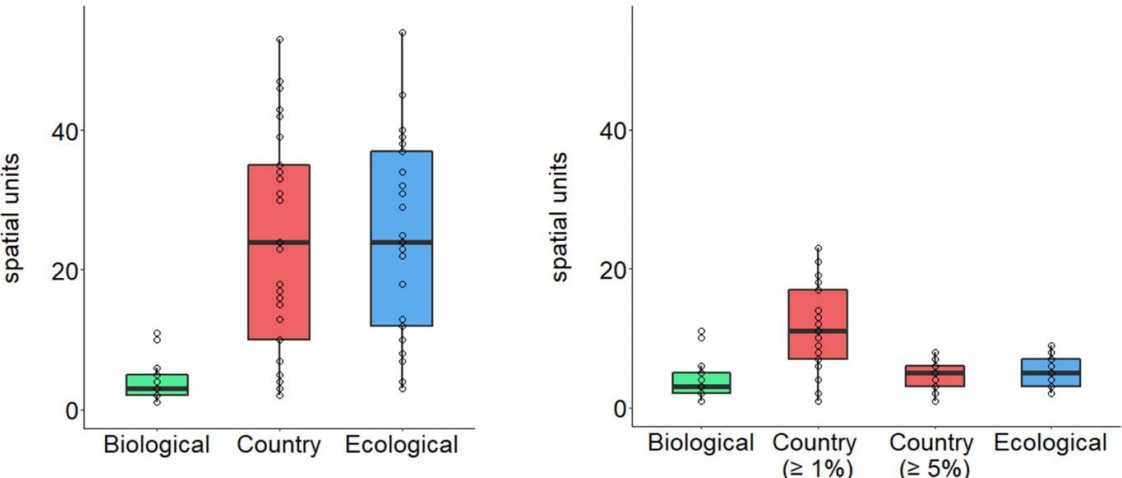

**Figure 1.** Boxplots showing the number of spatial units generated for 29 species of European mammals and birds using different methods of dividing the baseline distribution into spatial units. Center bars show median values. **Left:** Number of spatial units resulting from biologically based delineation methods compared with fine alternative spatial unit delineation methods (all countries within the baseline distribution treated as a separate spatial unit and ecological spatial units based on ecoregions). **Right:** Number of spatial units resulting from biologically based delineation methods compared with coarser alternative spatial unit delineation methods (only countries comprising ≥1% or ≥5% of the baseline distribution area treated as spatial units, and ecological spatial units based on biomes).

*3.2. Generation of Spatial Units too Small to Hold a Minimum Viable Population*

The use of fine-scale spatial unit delineation methods resulted in a larger number of spatial units that were estimated to be too small to hold a minimum viable population of 1000 mature individuals even under the best possible scenario (Figure 2). This was estimated by projecting the average density of mature individuals over the area (km²) of the spatial unit; if the total did not exceed 1000 mature individuals, the spatial unit was considered too small to ever be Viable. Fine-scale methods resulted in a greater proportion of such spatial units for both country-based methods (all countries, proportion of non-viable SUs = 40.7%, sd = 34.1%; countries ≥ 1% of baseline, proportion of non-viable SUs = 23.1%, sd = 34.7%; countries ≥ 5% of baseline, proportion of non-viable SUs = 12.6%, sd = 30.3%) and ecologically based methods (ecoregions, proportion of non-viable SUs = 48.8%, sd = 33.2%; biomes, proportion of non-viable SUs = 26.4%, sd = 33.4%).

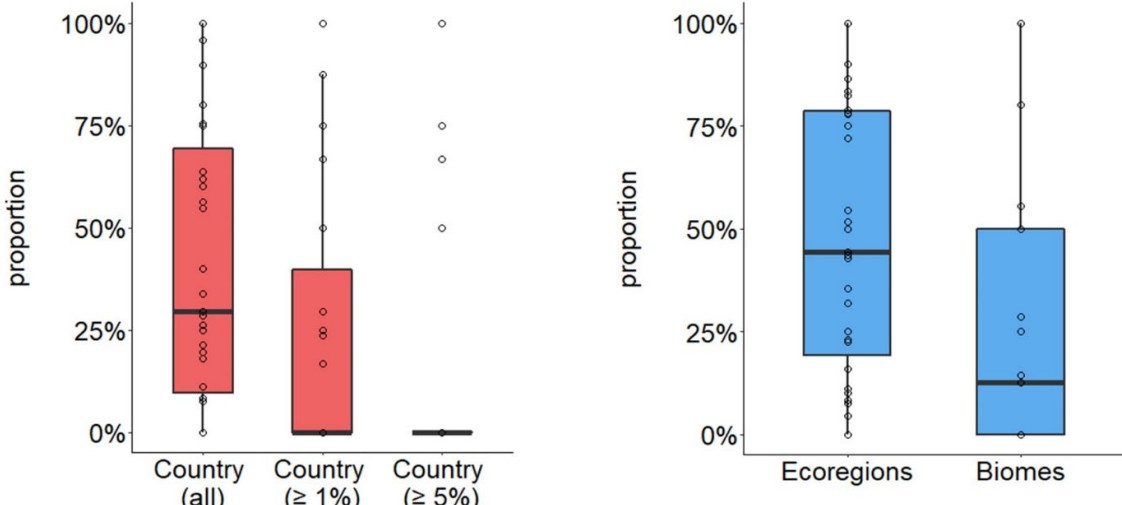

**Figure 2.** Boxplots showing the proportion of spatial units generated by a delineation method that were estimated to be too small to hold a minimum viable population of 1000 mature individuals, based on projecting the average density of mature individuals over the area (km²) of the spatial unit. **Left:** Proportion of spatial units that could never be considered Viable generated using countries as spatial unit delineations: fine divisions (all countries treated as separate spatial units); coarse divisions, ≥1% (all countries comprising at least 1% of the area of the baseline range treated as spatial units); and coarse divisions, ≥5% (all countries comprising at least 5% of the area of the baseline range treated as spatial units). **Right:** Proportion of spatial units that could never be considered Viable generated using ecologically based spatial units: finer divisions (ecoregions used as spatial units) and coarser divisions (biomes used as spatial units).

### 3.3. Average Changes in Green Score Based on Spatial Unit Delineation Method

When species' baseline distributions were divided into country- and ecoregion-based spatial units using fine divisions, the resulting Green Scores were significantly lower, on average, than the Green Scores generated using biologically based spatial units (Figure 3; BETAREG: log-likelihood = 39.3 on 4 df, $R^2$ = 0.12; country SUs average = 38%, sd = 16%, cf. biologically based SUs $p$ = 0.002; ecoregion SUs average = 39%, sd = 15%, cf. biologically based SUs $p$ = 0.005; biologically based SUs average = 51%, sd = 17%). However, when coarse divisions were used, there was no significant difference in the average Green Scores generated using those methods compared with the Green Scores generated using biologically based methods (Figure 3; BETAREG: log-likelihood = 49.2 on 5 df, $R^2$ = 0.020; country SUs (≥1%) average = 46%, sd = 18%, contrast with biologically based $p$ = 0.20; country SUs (≥5%) average = 52%, sd = 16%, contrast with biologically based $p$ = 0.96; ecoregion average = 50%, sd = 13%, contrast with biologically based $p$ = 0.75; biologically based SUs average = 51%, sd = 17%). Taxon was not a significant predictor in either model.

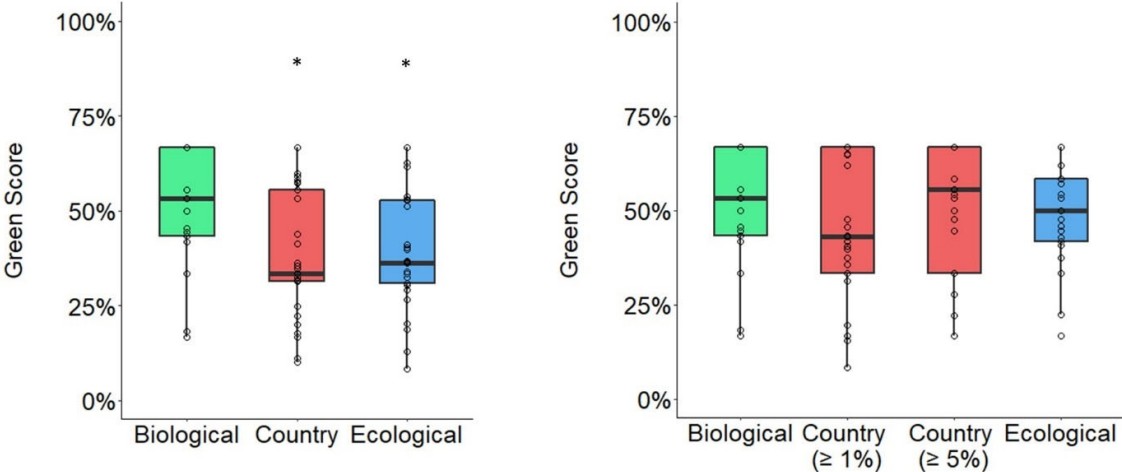

**Figure 3.** Boxplots showing Green Scores for 29 species of European mammals and birds generated using different methods of dividing the baseline distribution into spatial units. Center bars show median values, while asterisks represent means significantly different from the mean for the biological method (beta regression; see text). **Left**: Green Scores from biologically based spatial units compared with finer alternative spatial unit delineation methods (all countries within the baseline distribution treated as a separate spatial unit and ecological spatial units based on ecoregions). **Right**: Green Scores from biologically based spatial units compared with coarser alternative spatial unit delineation methods (only countries comprising ≥1% or ≥5% of the baseline distribution area treated as spatial units, and ecological spatial units based on biomes).

### 3.4. Species-Level Changes in Green Score Based on Spatial Unit Delineation Method

Looking at changes at the level of individual species, interesting patterns emerge. When considering changes in Green Score as a function of changes in number of spatial units for each species (Figures 4 and 5), the idea that fine-scale spatial unit delineation methods consistently generate more spatial units and lower Green Scores is reconfirmed for both ecologically based methods (Figure 4) and country-based methods (Figure 5). However, using coarse-scale methods, there is no longer a consistent pattern of difference; while fine-scale methods that produce more spatial units tend to correspond with lower Green Scores, when coarser-scale methods are used, country-based and ecologically based methods sometimes produce more, fewer, or the same number of spatial units as biologically based methods. This provides insight into the conditions under which the use of ecologically based or country-based methods is more or less appropriate, which we explored further using species-level case studies.

The results for each species generated using the coarse-scale methods are reported in Table 1.

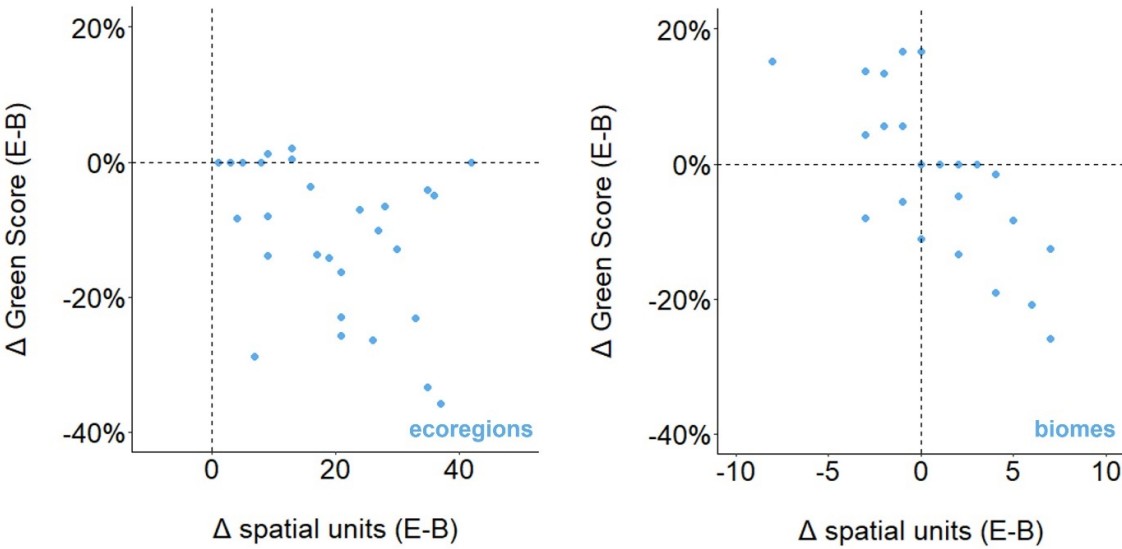

**Figure 4.** Change in Green Score as a function of change in number of spatial units when comparing ecologically based spatial units (SUs) with biologically based SUs (ecologically based results minus biologically based results; E-B). Each point represents a species (*n* = 29). Dashed lines intersect at the origin. Note the different x-axes. **Left**: Comparison of results between fine-scale ecologically based spatial units (ecoregions) and biologically based spatial units. **Right**: Coarse-scale ecologically based spatial units (biomes) compared to biologically based spatial units. Negative values on either axis indicate lower values were generated using ecologically based SUs compared to biologically based SUs.

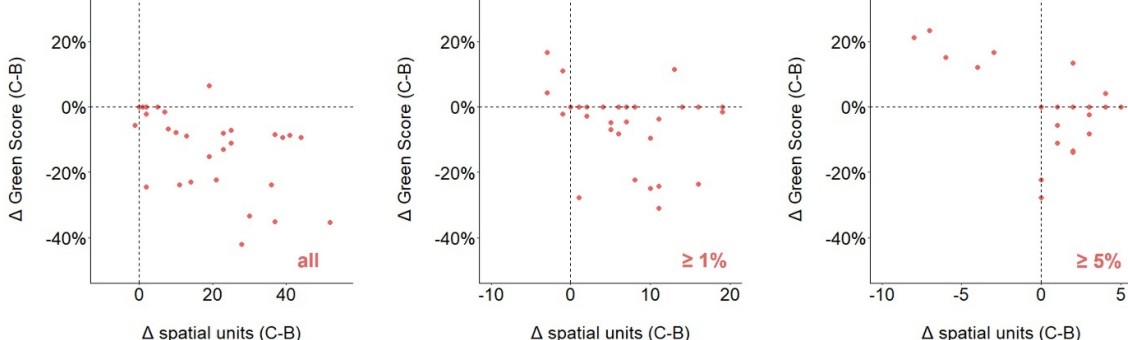

**Figure 5.** Change in Green Score as a function of change in number of spatial units when comparing country-based spatial units (SUs) to biologically based SUs (country-based results minus biologically based results; C-B). Each point represents a species (*n* = 29). Dashed lines intersect at the origin. Note the different x-axes. **Left**: Comparison of results between fine-scale country-based spatial units (all countries within baseline distribution) and biologically based spatial units. **Center**: Coarse-scale country-based spatial units (countries comprising ≥1% of baseline distribution) compared to biologically based spatial units. **Right**: Coarse-scale country-based spatial units (countries comprising ≥ 5% of baseline distribution) compared to biologically based spatial units. Negative values on either axis indicate lower values were generated using ecologically based SUs compared to biologically based SUs.

### 3.5. Ecological vs. Biological Case Studies

For some species, the outcomes generated using ecologically based spatial units were identical to those generated using biologically based spatial units (i.e., the species located at the origins of Figure 4). When biomes were used, the resulting Green Scores and number of spatial units were the same as the results for biologically based spatial units for two species: the cinereous vulture *Aegypius monachus* and the griffon vulture *Gyps fulvus* (Table 1). For both species, the baseline range was semi-contiguous and large (>800,000 km²).

Although coarser-scale ecologically based methods of delineating spatial units (biomes) were generally more consistent with biologically based methods than finer-scale methods, this was not true for all species. In some cases, using biomes as spatial units resulted in assignment of fewer spatial units, and a higher Green Score, than biologically based spatial units. This was the case for the bearded vulture *Gypaetus barbatus*. The species has a discontinuous baseline range, producing 10 biologically based spatial units. However, when spatial units were based on biomes, subpopulations occurring across a biome were considered part of the same spatial unit (even though they were discontinuous), resulting in fewer spatial units (Figure 6). Because geographically separate areas were lumped together using the biome method, and their mature individuals counted together to gauge viability, the Green Score was overestimated.

In some cases, using biomes as spatial units resulted in the assignment of more spatial units, and a lower Green Score, than biologically based spatial units. This was the case for the white-tailed eagle *Haliaeetus albicilla* (Figure 7). The species has a continuous baseline distribution, divided into two biologically based spatial units based on subpopulations (a larger northeast subpopulation and smaller southern population, both Viable). In contrast, the distribution covers nine biomes. Due to the large and continuous baseline range, areas of distribution that fall within a single biome are less fragmented than those in the case of *Gypaetus barbatus*. However, the relative size of biomes is more variable, and the distribution is made up mostly of two of the nine, while some biomes were not large enough to hold a minimum viable population (Figure 7). Given the relatively consistent spatial distribution of individuals across the species' range, if spatial units were chosen to be large enough to hold a viable population, ecoregions would likely have produced outcomes that were the same as or similar to those of biologically based spatial units.

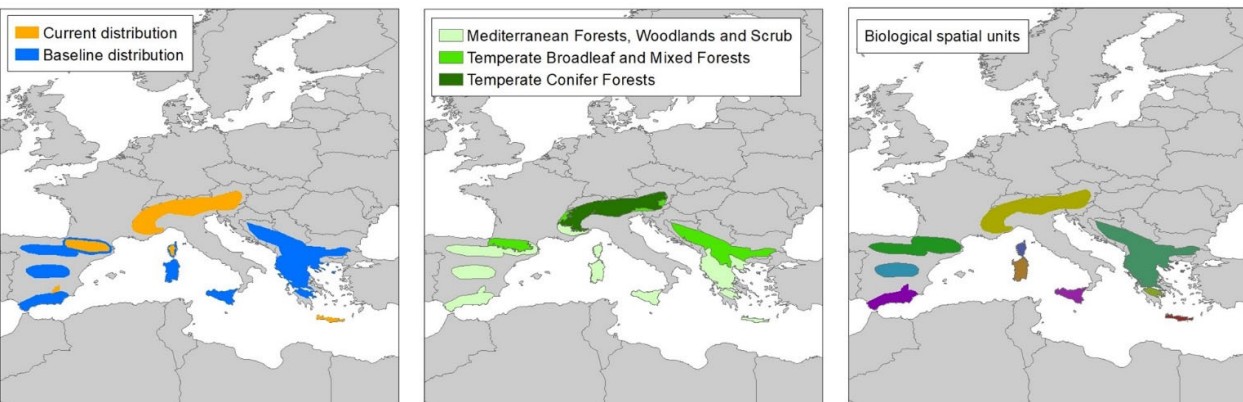

**Figure 6.** Spatial units based on biomes (coarse-scale) overestimate Green Score compared with biologically based spatial units for the bearded vulture *Gypaetus barbatus*. **Left**: The current distribution covers less than half of the baseline distribution. **Center**: Dividing the baseline distribution by biome results in three spatial units, but these units are discontinuous (all areas with same color considered same spatial unit). The species is therefore considered Present in all three spatial units. **Right**: Using biologically based spatial units results in 10 spatial units, none of which are discontinuous; the species is considered Present in half of them.

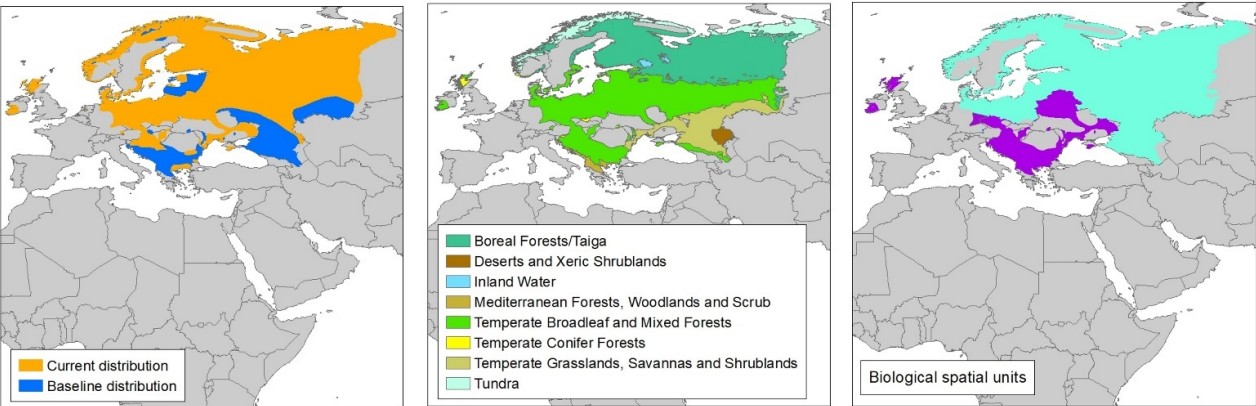

**Figure 7.** Spatial units based on biomes (coarse-scale) underestimate Green Score compared with biologically based spatial units for the white-tailed eagle *Haliaeetus albicilla*. **Left**: The current distribution covers the majority of the baseline distribution. Note that the baseline includes western Iceland, and the species is present there, but Iceland is not mapped due to space restrictions. **Center**: Dividing the baseline distribution by biome results in nine spatial units, although two biomes cover the majority of the distribution (all areas with same color considered same spatial unit). The species is considered Viable in these two spatial units and Present or Absent in the rest. Note that Iceland (not mapped) contains an additional small "Rock and Ice" spatial unit. **Right**: Using biologically based spatial units results in two spatial units, and the species is Viable in both.

*3.6. Country vs. Biological Case Studies*

As above, there were cases where spatial units based on countries produced the same result as biologically based spatial units (i.e., the species located at the origins of Figure 5). When the coarse spatial units (≥5%) were used, the same Green Score and number of spatial units were generated as for biologically based spatial units for three species: the eastern imperial eagle *Aquila heliaca*, the Spanish imperial eagle *Aquila adalberti*, and the pink-footed goose *Anser brachyrhynchus* (Table 1). For these species, the different biological spatial units occurred either entirely within a country or extended only slightly outside of a country (meaning that spatial units based on countries that comprised ≥ 5% of the baseline range matched with the biological spatial units).

Even after excluding countries that made up less than 5% of the baseline range, there were cases where using countries as spatial units resulted in the assignment of more spatial units, and a lower Green Score, than biologically based spatial units. The species that shows this most clearly is the cinereous vulture *Aegypius monachus* (Table 1). In this case, multiple countries occurred within an area of dispersal for the species (Figure 6). Conversely, there were cases where using countries as spatial units overestimated the Green Score relative to biologically based spatial units. This is well demonstrated by the Iberian ibex *Capra pyrenaica*, which is primarily restricted to Spain (<5% of baseline range occurs in Portugal); therefore, only one spatial unit was assigned using the country method. However, within Spain, three subspecies existed in the baseline year (two currently extant, one of which is threatened). Using country spatial units, these Absent, Present, and Viable subspecies were collapsed into a single Viable population.

*3.7. Cases Consistent across Spatial Unit Delineation Methods*

There were seven species where the same Green Score was generated using all methods of delineating spatial units—even when ecological or country-level divisions produced more spatial units than biologically based divisions: Spanish imperial eagle *Aquila adalberti*, pink-footed goose *Anser brachyrhynchus*, red deer *Cervus elaphus*, white stork *Ciconia ciconia*, wolverine *Gulo gulo*, white-headed duck *Oxyura leucocephala*, and wild boar *Sus scrofa* (Table 1). These species shared the following characteristics: the vast majority of their baseline distribution was still occupied (or re-occupied) in 2013, and their threat

status was consistent across the range—either threatened in all areas or Least Concern in all areas.

## 4. Discussion

Although the preferred method of delineating spatial units for a Green Status of Species assessment is to use existing biological divisions such as subpopulations [1,8], information on such divisions is not always available; in these cases, a different method for delineating spatial units must be used. We identified potential pitfalls that assessors could encounter when doing so by comparing Green Scores generated using biologically based spatial units to outcomes generated for the same species using two alternative methods: ecologically based and country-based spatial units.

### 4.1. Findings and Recommendations

The first key finding is that consideration of connectivity is crucial when delineating spatial units. The focal area for this study was Europe, which comprises a large number of relatively small countries whose political borders are usually not a barrier to species movement. When fine-scale country-level methods were used, the spatial units included marginal (relative to the core distribution) and small countries as individual spatial units, even though they made up very little area of the baseline distribution; often these spatial units were too small to ever become Viable (and therefore, in a full Green Status assessment, Functional). Similarly, when using fine-scale ecological spatial units, ecoregions showing fine-scale variation in characteristics were treated as separate spatial units, even though this variation was likely not relevant for many species (e.g., separation of two different "Mixed Forests" ecoregions [12] into two spatial units, even though they are contiguous with each other). Although no minimum size of spatial units is specified in the Green Status guidance, it does state that spatial units should "represent areas of similar importance for the species' conservation" [1]. The importance of taking this into account is underlined by our result that using finer-scale ecologically based and country-based spatial units resulted in significantly lower Green Scores than using biological spatial units. Therefore, assessors should avoid using ecologically based spatial units, and especially country-based spatial units, unless there is reason to believe, based on the biology of the species, that these approximately correspond to biological units (i.e., subpopulations), and that each one is large enough to theoretically hold a Viable and Functional population.

The need to delineate spatial units so that they could potentially hold a Viable and Functional population is not trivial; a central concept of the Green Status of Species is that if a species were restored to pre-impact levels in all of its spatial units, it would be fully recovered, i.e., Viable and Functional across its range. If spatial units are designed in such a way that they could not ever be expected to hold a viable population, then the recovery potential of the species is artificially reduced. It is true that for some endemic species occurring at low densities, even treating the entire range as a single spatial unit might not have the potential to reach a minimum viable population; exceptions for such species are described in the Green Status guidance materials [1,8]. Nonetheless, in most cases, the realization that some spatial units are too small to hold a minimum viable population should cause assessors to reassess the scale and connectivity of their chosen spatial units.

Coarse-scale methods for generating ecologically based and country-based spatial units performed better overall for our European focal species, generating much lower proportions of spatial units that could not hold a minimum viable population (Figure 2) and eliminating the difference in number of spatial units and in Green Scores when averaged across species (Figures 1 and 2). However, the individual species results demonstrated that connectivity must still be a key consideration. This was demonstrated most profoundly when using biomes as spatial units; in many cases, this resulted in more spatial units than the biological method because connected biomes, within the dispersal limits of the species and acting as a single biological unit, were broken into multiple ecological spatial units (e.g., Figure 7). Therefore, to avoid artificially depressing the Green Score

when using ecologically based spatial units, assessors should take connectivity of ecologically distinct regions into account and, if necessary, collapse them into larger spatial units that reflect the characteristics of the species.

The use of biomes as spatial units sometimes artificially inflated the Green Score as well. In species with multiple biological spatial units classified as the same biome, these biological spatial units were collapsed into just one spatial unit using the coarse-scale ecological method, even if they were geographically separated (e.g., Figure 6). For species that have been extirpated in parts of their range, such lumping based on ecoregions hides the reality of range loss. Therefore, it is important that spatial units are chosen to reflect the representation of the species across the range [1]; if it has been extirpated in much of its range, this should be reflected in the choice of spatial units. For species where ecoregions are not contiguous across the species distribution, the ecoregions should not, in most cases, be aggregated together. We recognize an area of potential confusion around this recommendation, as the Green Status guidelines state that "a species' distribution within a spatial unit does not have to be contiguous" [8]. However, the current guidance refers primarily to a distribution that was connected prior to human impact that has since become fragmented, not to similar ecological areas that have historically been separated by another ecoregion or biome.

Using countries as spatial units presents conceptually similar potential pitfalls. Political borders present even less of a barrier to connectivity than ecological borders, so if part of a species' distribution crosses political borders, and the species is faring similarly in the different countries, those countries should likely be collapsed into a single spatial unit to better reflect biological reality. Conversely, if a species occurs primarily within a single country, assessors should consider whether the status of the species varies across the country, and how this can be reflected in the Green Score by the choice of spatial units.

For species with certain characteristics, the choice of spatial units appears to matter less; for such species, the same Green Score was generated across the three methods, even when those methods divided the range up in very different ways (Table 1). This result occurred for species faring uniformly across their baseline range—if the species has not been extirpated anywhere, and is uniformly threatened or non-threatened, then as long as spatial units are large enough to hold a Viable population, the Green Score will be the same no matter how spatial units are produced. However, the Green Status aims not only to measure a species' *current* recovery status, but also to incentivize conservation by projecting how recovery status might change in the future. Therefore, even if a species' status is uniform across its range at the time of assessment, the spatial units need to be able to capture future changes in status that may happen non-uniformly across the range due to spatial variation in threats and conservation actions.

### 4.2. Scope of Results and Future Directions

While this analysis has provided key insights into the trade-offs between the different methods of delineating spatial units, it has a number of limitations. For one, the states of our country-based and ecologically based spatial units were not based on real-world data, but generated by projecting a uniform density of individuals across the occupied areas of the baseline distribution; in reality, such homogeneity across the range is unlikely. While we were able to determine the extent of this discrepancy for spatial units based on countries (~85% consistency with country-level statistics we could glean from [10]), we were unable to do so for ecoregions or biomes. Therefore, it is possible that ecologically based Green Scores were over- or underestimated for some species.

When considering coarser-scale methods for countries, we excluded countries that fell beyond the specified thresholds (at least 1% or 5% of the baseline distribution area). This was appropriate for this exercise as it prevented peripheral countries from artificially deflating the Green Score, but in a true Green Status assessment, all parts of the indigenous range must be included in the delineation of spatial units. In a real assessment, these peripheral

countries could instead be considered part of a larger spatial unit, made up of many connected countries. Our results approximate this approach but were based on a simplification.

Beyond the exclusion of some parts of the range in coarser-scale country analyses, we used highly modified Green Status of Species assessment methods overall to accommodate the constraints of the analysis. Firstly, we did not determine the indigenous range for our focal species, rather using their distribution in the recent past as a baseline distribution for delineating spatial units and calculating Green Scores. Secondly, we used greatly simplified criteria for determining the viability of a spatial unit, comparing population size to a simple threshold of 1000 mature individuals rather than applying the Red List Criteria using the regional guidelines [7,15], which were designed to guard against oversimplification. This means that some spatial units that were considered Viable in this exercise would not be in a full assessment.

Finally, we were not able to incorporate functionality, a key part of the Green Status assessment, in this exercise. If we could assess functionality, it is possible that we would find more differences between country-based or ecologically based spatial units, and more advantages of the latter. Ecological divisions such as ecoregions, habitat types, or ecosystems can be used to define spatial units because they capture the different "ecological settings" in which a species exists (or has existed), therefore encapsulating the different ecological roles and functions of the species. Assessing functionality across the species range would uncover these differences and may highlight the relevance of ecologically based spatial units.

Despite these limitations, our methods allowed us to identify key cases where methods for generating country-based or ecologically based spatial units either perform well or fall short. Nonetheless, the results reported here for the focal species should not be taken as the actual Green Status assessments for these species.

This study explored three methods of delineating spatial units for a Green Status assessment, but it was not exhaustive. For example, when considering anthropogenic boundaries, we used countries given the European context of the exercise; however, in other regions, different political boundaries may be more appropriate, e.g., states, counties, or provinces. We also did not explore the use of delineating spatial units using grid cells of varying sizes, or of using levels to create spatial units (e.g., subpopulations further subdivided by ecoregion) [8]. Nonetheless, our results should inform the delineation of spatial units using levels, given that our findings highlight the importance of spatial units that capture relevant variation in status or connectivity.

A lack of data limited this exercise to a small set of species, taxonomically restricted to birds and mammals, ecologically restricted to terrestrial species, and geographically restricted to Europe. Although this provided useful initial insights to guide delineation of spatial units, which likely apply across many contexts, further testing with freshwater and marine species, as well as terrestrial plants, fungi, invertebrates, amphibians, and reptiles occurring in varied geographies, will likely uncover unique cases and raise different considerations to further guide assessors wishing to undertake Green Status of Species assessments.

**Supplementary Materials:** The following supporting information can be downloaded at: https://www.mdpi.com/article/10.3390/d14090742/s1, Table S1: Information used to derive Green Scores using biologically based spatial units; Table S2: Correction factors used to estimate number of mature individuals from total reported population size; Table S3: Comparison of projected country-level population sizes to reported country-level population sizes; Table S4: Justifications for state estimations in biologically based spatial units [18–48].

**Author Contributions:** Conceptualization, M.K.G.; M.J.W.B.; A.P.; D.M.; H.R.A.; methodology, M.J.W.B.; M.K.G.; software, M.J.W.B.; validation, M.J.W.B.; M.K.G.; formal analysis, M.K.G.; M.J.W.B.; data curation, M.J.W.B.; M.K.G.; writing-original draft preparation, M.K.G.; M.J.W.B.; writing-review and editing, M.K.G.; H.R.A.; E.L.B.; M.J.W.B.; C.H.-T.; M.H.; D.M.; A.P.; R.Y. (Rebecca Young), R.Y (Richard Young)., B.L.; visualization, M.J.W.B.; M.K.G.; supervision, M.K.G.; A.P.; D.M.; project administration, C.H.-T.; funding acquisition, B.L.; C.H.-T.; M.H.; M.K.G.; R.Y Richard Young)., E.L.B. All authors have read and agreed to the published version of the manuscript.

**Funding:** This research was funded by Prince Albert II of Monaco Foundation, grant number CCI-05-19-008 administered via the Cambridge Conservation Initiative Collaborative Fund, and by the Natural Environment Research Council, grant number NE/S006125/1. The APC was funded by the Open Access Oxford block grant.

**Data Availability Statement:** The species distributions and population data used in this article are available in the following publicly available report: Deinet, S.; Ieronymidou, C.; McRae, L.; Burfield, I.J.; Foppen, R.P.; Collen, B. and Böhm, M. (2013) Wildlife comeback in Europe: The recovery of selected mammal and bird species. Final report to Rewilding Europe by ZSL, BirdLife International and the European Bird Census Council. London, UK: ZSL. https://rewildingeurope.com/wp-content/uploads/2013/11/Wildlife-Comeback-in-Europe-the-recovery-of-selected-mammal-and-bird-species.pdf (accessed on 16 August 2022).

**Acknowledgments:** Thanks to Monika Böhm and Ian Burfield, who provided access to the species map shapefiles from Deinet et al. (2013).

**Conflicts of Interest:** The authors declare no conflict of interest. The views expressed in this publication do not necessarily reflect those of IUCN. The designation of geographical entities in this paper, and the presentation of the material, do not imply the expression of any opinion whatsoever on the part of IUCN concerning the legal status of any country, territory, or area, or of its authorities, or concerning the delimitation of its frontiers or boundaries.

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
