# Peer review of "The Impact of Spatial Delineation on the Assessment of Species Recovery Outcomes"

_diversity, doi:10.3390/d14090742_

Round 1
Reviewer 1 Report
Review of diversity-1892608
The present manuscript consists of a meta-analysis investigating the effects of different scales of range division, the ‘spatial units’, for evaluating the IUCN Green Score. The authors compare the assessment calculated based on the recommended divisions, the biologically-based spatial units, to alternative divisions, namely ecologically-based spatial units and politically-based spatial units, i.e. countries. Moreover, within each of the alternative approaches, the authors further tested for fine and for coarse categories to assess their strength. This comparison was performed on the real distribution of 15 bird and 14 mammal species in Europe.
The authors discuss adequately the pros and cons of each method and provide suggestions for their use according to species-specific criteria and acknowledge the limitations of the study based on their methodology. This manuscript should be considered as a useful guideline for the use of the different methods in evaluating the IUCN Green Score. It is very well written and provides insights for future work towards this end.
The methods are well explained and are well detailed. The analyses are thorough, and the authors have followed the IUCN criteria almost religiously. However, the fact that the authors could not classify any divisions in the Functional category poses some problems, but it is overcome by acknowledging the underestimation of the Green score.
In general, the use of fine divisions, resulted into many more spatial units, on average, using the country- and ecologically-based divisions than the biologically-based divisions. This was consistent for all studied species, irrespective of their variable biology. The Green Scores deriving from the country- and ecologically-based divisions were significantly lower than the Green Scores generated using the biologically-based spatial units. In contrast, the use of coarser divisions, produced similar average number of spatial units in country-, ecologically-, and biologically-based methods. Herein, there was no consistent pattern of difference regarding the studied species. There was no significant difference in the average Green Scores deriving from all those three methods.
In subsection 3.7, it would be easier for the reader that the authors name the seven species in the text, rather than referring to Table 1. This would keep consistency with the previous subsections of the Results.
The discussion is well presented and analyzes in detail the potential of the different methods, underlines their limitations, and provides insight for future prospects.
Author Response
Review comment 1:
The present manuscript consists of a meta-analysis investigating the effects of different scales of range division, the ‘spatial units’, for evaluating the IUCN Green Score. The authors compare the assessment calculated based on the recommended divisions, the biologically-based spatial units, to alternative divisions, namely ecologically-based spatial units and politically-based spatial units, i.e. countries. Moreover, within each of the alternative approaches, the authors further tested for fine and for coarse categories to assess their strength. This comparison was performed on the real distribution of 15 bird and 14 mammal species in Europe.
The authors discuss adequately the pros and cons of each method and provide suggestions for their use according to species-specific criteria and acknowledge the limitations of the study based on their methodology. This manuscript should be considered as a useful guideline for the use of the different methods in evaluating the IUCN Green Score. It is very well written and provides insights for future work towards this end.
The methods are well explained and are well detailed. The analyses are thorough, and the authors have followed the IUCN criteria almost religiously. However, the fact that the authors could not classify any divisions in the Functional category poses some problems, but it is overcome by acknowledging the underestimation of the Green score.
In general, the use of fine divisions, resulted into many more spatial units, on average, using the country- and ecologically-based divisions than the biologically-based divisions. This was consistent for all studied species, irrespective of their variable biology. The Green Scores deriving from the country- and ecologically-based divisions were significantly lower than the Green Scores generated using the biologically-based spatial units. In contrast, the use of coarser divisions, produced similar average number of spatial units in country-, ecologically-, and biologically-based methods. Herein, there was no consistent pattern of difference regarding the studied species. There was no significant difference in the average Green Scores deriving from all those three methods.
Authors' response 1:
We thank Reviewer 1 for their positive comments and for the thorough and accurate summary of our findings.
Review comment 2:
In subsection 3.7, it would be easier for the reader that the authors name the seven species in the text, rather than referring to Table 1. This would keep consistency with the previous subsections of the Results.
Authors' response 2:
We thank Reviewer 1 for this helpful suggestion. We have listed the species' common and scientific names in the text before referring to Table 1.
Review comment 3:
The discussion is well presented and analyzes in detail the potential of the different methods, underlines their limitations, and provides insight for future prospects.
Authors' response 3:
We thank Reviewer 1 for this positive comment.
Reviewer 2 Report
The manuscript reports an excellent study in its conception, execution, and presentation and discussion of the results. The manuscript is very well written, and the methods and results supported by excellent tables and figures including supplementary tables as appropriate. I had some concerns about the generality of the results at the spatial level of country given endemism in large countries like the USA and Australia. However, the authors ably addressed this concern (lines 715-17) in the discussion and rightly raised the use of states, provinces and territories as political divisions of large jurisdictions akin to the European countries modelled.
I found figures 6 and 7 compelling in understanding the relationship between species distribution and the spatial units used in the study. I thought to suggest a figure of this kind in the methods, but these are probably sufficient.
There is a formatting error at line 488 where Table 1 heading is not separated from Figure 5. At line 558 species is misspelled.
My one concern is Table 1. The entries are unsorted and make it unnecessarily difficult to follow. The text highlights the difference values of GS ecology and to a lesser extent GS country along with the baseline distribution. I suggest sorting the lines on column 6 and 9 to simplify the interpretation of this table.
Author Response
Review comment 1:
The manuscript reports an excellent study in its conception, execution, and presentation and discussion of the results. The manuscript is very well written, and the methods and results supported by excellent tables and figures including supplementary tables as appropriate. I had some concerns about the generality of the results at the spatial level of country given endemism in large countries like the USA and Australia. However, the authors ably addressed this concern (lines 715-17) in the discussion and rightly raised the use of states, provinces and territories as political divisions of large jurisdictions akin to the European countries modelled.
I found figures 6 and 7 compelling in understanding the relationship between species distribution and the spatial units used in the study. I thought to suggest a figure of this kind in the methods, but these are probably sufficient.
Authors' response 1:
We thank Reviewer 2 for their positive and thorough review of our paper, its findings, and presentation.
Review comment 2:
There is a formatting error at line 488 where Table 1 heading is not separated from Figure 5.
Authors' response 2:
Figure 5 and Table 1 have now been separated with a blank line (this now occurs at line 403, as we have adjusted the indent of the paper, which via (presumably) an upload error was altered to be very deep, adding many extra lines to the ms.
Review Comment 3:
At line 558 species is misspelled.
Authors' response 3:
Thank you, this has been corrected (now at line 461)
Review Comment 4:
My one concern is Table 1. The entries are unsorted and make it unnecessarily difficult to follow. The text highlights the difference values of GS ecology and to a lesser extent GS country along with the baseline distribution. I suggest sorting the lines on column 6 and 9 to simplify the interpretation of this table.
Authors' response 4:
We have gone back and forth on how to present this table- thank you for your suggestion! In the submission, the table was sorted by the number of biologically-based SUs, as it was proving tricky to sort another way given that there are 4 columns of interest (change in SUs for ecoregion- and country-based methods; change in Green Score for ecoregion- and country-based methods). However, we agree with Reviewer 2 that the most relevant information is the change in Green Score produced using the 2 other methods (columns 6 and 8; we assume based on context that the Reviewer's mention of column 9 was a typo).
We have therefore rearranged the table so that the species for which all methods yielded the same Green Score are presented first, followed by species where one other method produced the same Green Score as the biological method, followed by species where neither alternative method produced the same Green Score. Gray shading has been added to assist in the visual division between these groups and improve the readability of the table.